# Therapeutic Consequences of Targeting the IGF-1/PI3K/AKT/FOXO3 Axis in Sarcopenia: A Narrative Review

**DOI:** 10.3390/cells12242787

**Published:** 2023-12-07

**Authors:** Benjamin Gellhaus, Kai O. Böker, Arndt F. Schilling, Dominik Saul

**Affiliations:** 1Department of Trauma, Orthopedics and Reconstructive Surgery, Georg-August University of Goettingen, 37075 Goettingen, Germany; benjamin.gellhaus@med.uni-goettingen.de (B.G.); kai.boeker@med.uni-goettingen.de (K.O.B.); arndt.schilling@med.uni-goettingen.de (A.F.S.); 2Department of Trauma and Reconstructive Surgery, Eberhard Karls University Tuebingen, BG Trauma Center Tuebingen, 72072 Tuebingen, Germany; 3Division of Endocrinology, Mayo Clinic, Rochester, MN 55905, USA; 4Robert and Arlene Kogod Center on Aging, Mayo Clinic, Rochester, MN 55905, USA

**Keywords:** sarcopenia, FOXO3, skeletal muscle atrophy, aging, satellite cells, Atrogin-1, Murf-1

## Abstract

The high prevalence of sarcopenia in an aging population has an underestimated impact on quality of life by increasing the risk of falls and subsequent hospitalization. Unfortunately, the application of the major established key therapeutic—physical activity—is challenging in the immobile and injured sarcopenic patient. Consequently, novel therapeutic directions are needed. The transcription factor Forkhead-Box-Protein O3 (FOXO3) may be an option, as it and its targets have been observed to be more highly expressed in sarcopenic muscle. In such catabolic situations, *Foxo3* induces the expression of two muscle specific ubiquitin ligases (*Atrogin-1* and *Murf-1*) via the PI3K/AKT pathway. In this review, we particularly evaluate the potential of *Foxo3*-targeted gene therapy. *Foxo3* knockdown has been shown to lead to increased muscle cross sectional area, through both the AKT-dependent and -independent pathways and the reduced impact on the two major downstream targets *Atrogin-1* and *Murf-1*. Moreover, a *Foxo3* reduction suppresses apoptosis, activates satellite cells, and initiates their differentiation into muscle cells. While this indicates a critical role in muscle regeneration, this mechanism might exhaust the stem cell pool, limiting its clinical applicability. As systemic *Foxo3* knockdown has also been associated with risks of inflammation and cancer progression, a muscle-specific approach would be necessary. In this review, we summarize the current knowledge on *Foxo3* and conceptualize a specific and targeted therapy that may circumvent the drawbacks of systemic *Foxo3* knockdown. This approach presumably would limit the side effects and enable an activity-independent positive impact on skeletal muscle.

## 1. Introduction

According to the definition by Rosenberg, sarcopenia is characterized as a loss of muscle mass and function due to aging [1]. The European Working Group on Sarcopenia in Older People (EWGSOP2) published an updated definition (2018) for sarcopenia based on three main criteria: 1. Lower muscle strength is accompanied by 2. an impaired muscle quality and quantity and 3. lowered physical performance in the patient. A probable sarcopenia is identified by the first criterion and the diagnosis is confirmed by the second. If all three criteria are met by a patient, sarcopenia is diagnosed as severe. Moreover, sarcopenia is subdivided by its etiology into primary (aging) and secondary (disease/inflammation, inactivity, malnutrition) [2]. Remarkably, the lowered muscle strength became the major criterion. In 2010, the EWGSOP defined sarcopenia on lower muscle mass as the first, lower muscle strength as the second, and low physical performance as the third criterion. Here, depending on how many criteria were met, the stages presarcopenia (only criterion one), sarcopenia, and severe sarcopenia (all criteria) were defined [3].

Furthermore, it is of crucial importance to differentiate sarcopenia from muscle atrophy and cancer cachexia. Atrophy is defined as cell shrinkage leading to a decreased size of a tissue or an organ. Regarding muscle atrophy, the skeletal muscle fiber size is decreased [4]. In contrast to that, cancer cachexia is defined as a loss of muscle mass due to deregulated energy metabolism that is almost resistant to nutritional supplementation [5]. Basically, atrophy is part of both definitions (cancer cachexia and sarcopenia), but cancer cachexia is defined via metabolic imbalances, while sarcopenia focuses on the loss of function.

## 2. Epidemiology and Economy

Among a population of 518 male European participants aged 40–79 years, an incidence of 1.6% was observed after a follow-up of 4.3 years, as defined by the EWGSOP criteria for sarcopenia [6]. Furthermore, another study (n = 719) conducted on older (≥85 years) female and male participants found a prevalence of 21% for sarcopenia based on the EWGSOP criteria with an incidence of 3.7% in a follow-up after 3 years [7]. A study investigating sarcopenia in older hospitalized patients (≥65 years) revealed 35% of the patients to be sarcopenic at hospital admission (prevalence) and 15% of non-sarcopenic patients to develop sarcopenia before discharge (incidence in hospitalized patients), indicating its relevance and tremendous impact on the healthcare system [8].

A follow-up study in geriatric patients (7.1% sarcopenic according to EWGSOP definition, n = 445) observed that during a period of three years, about half of the patients (≥65 years) fell one or multiple times, while the fall rate was higher in sarcopenic patients [9]. The increased rate of falling was confirmed in an extensive meta-analysis (n = 45,926), in which a positive correlation of falls with fractures was observed in sarcopenic patients [10]. Intriguingly, grip strength, as a major criterion used to identify sarcopenia, but also walking speed, chair raises, and standing balance were found to be associated with mortality in a meta-analysis. Individuals with lower performance in these metrics were found to have a greater risk of mortality, highlighting the importance of parameters used to identify sarcopenia and risk of death [11]. In addition, hospital stays are prolonged in sarcopenic compared to non-sarcopenic patients, which has an economic impact on the health system [12]. In detail, prolonged hospital stays lead to a higher socioeconomic burden in the healthcare system, resulting in an estimated additional cost of $40.4 billion for patients with sarcopenia in the US in 2014 [13].

## 3. Current Therapeutic Concepts in Sarcopenia

It has been extensively reviewed that age-related changes in muscle structure occur both quantitatively and qualitatively. The progressive loss of motor unit innervation and impaired reinnervation (quantitative) coupled with molecular protein turnover (qualitative) lead to reduced muscle function [14]. To regain muscle function, modern treatment of sarcopenia is based on two major principles: physical activity and oral supplementation [15].

A study in older male and female participants (70–90 years, n = 1635) with physical limitation (short physical performance battery [SPPB] ≤ 9) participated in moderate physical activity training (intervention) or a health education program (control). In a follow-up after 2.6 years, the incidence of major mobility disability (400 m walk within 15 min) was significantly reduced in the intervention group [16]. Another study (n = 124, follow-up for 12 months) focused on high-intensity weightlifting training compared to a multidisciplinary intervention in hip-fractured patients. The authors revealed that mortality and nursing home admission were reduced by 81% and 84%, respectively [17]. Both concepts seem to improve the endpoints but require long-lasting therapy that is not suitable for every sarcopenic patient.

In particular for hospitalized patients, it might be specifically challenging, since over 80% of the time spent in the hospital is bedridden (≥65 years, n = 45) [18]. Incidentally, immobilization was found to increase *Foxo3* acetylation in vivo, in mice with unilateral hindlimb immobilization, which will be further discussed in the following paragraph [19]. Consequently, other forms of treatment need to be explored.

The first option to improve sarcopenia is the optimization of nutrition. To maintain and regain new body mass, the PROT-AGE working group recommends 1.0–1.2 g/kg BW of protein intake per day for older people (>65 years) [20]. Nutritional status as a key for maintaining muscle mass was investigated in a prospective randomized controlled trial in older (≥65 years) sarcopenic subjects receiving a supplementation therapy (protein-, leucine, vitamin D-, and mineral-enriched) for 13 weeks, resulting in increased appendicular lean mass but no differences in physical performance and strength [21].

Another similar study focused on immobilization. In this trial, older (>65 years) hip-fractured participants received hypercaloric and protein-enriched supplementation. An increase in appendicular lean mass was observed from admission to discharge, whereas weight and muscle mass were kept at a constant level, preventing a decrease in muscle mass compared to subjects receiving a standard diet [22]. Both studies point out the importance of nutrition in sarcopenic patients by keeping weight and muscle mass constant even if functional differences could not be observed.

Vitamin D in combination with calcium was identified to reduce the risk of falling in a huge meta-analysis (n = 45,782). This effect was mainly attributed to patients having a vitamin D deficiency [23], but depending on the location, this can be the majority of the population [24]. Further, another meta-analysis (n = 5615) revealed a positive effect of (various doses of) vitamin D on muscle strength but not on muscle mass and power in older subjects (61 years) [25]. Surprisingly, no effect on all-cause mortality could be detected in a large meta-analysis (n = 74,655), making Vitamin D a controversial treatment that has been followed up sophistically [26,27]. Another therapeutic option for sarcopenia is hormone supplementation, especially with testosterone. The daily application of 1% testosterone gel over 12 months in male subjects (≥65 years, n = 790) resulted in a better 6-min walk distance [28]. Moreover, its application for 36 months, also in male subjects (≥60 years, n = 256), led to increased stair-climbing power, muscle strength, and lean body mass [29]. Clinically, only a few side effects were reported for testosterone gel. Besides skin irritation in some patients, removing by washing may limit the applicability. But it is of importance to notice that these side effects were described for supplementation to serum testosterone levels within the physiological range. Hence, an increase in prostate-specific antigen could not be verified [30].

Thus, new therapeutical avenues are required. The objective of this review is to open new therapeutical areas of a *Foxo3*-targeted treatment in sarcopenia and give an insight into their underlying molecular mechanisms.

## 4. The Forkhead Box Family and Its Link to Sarcopenia

Out of many controlling genes within the musculoskeletal system, Forkhead-Box Protein O3 (FOXO3) is pivotal due to its inactivation during muscle growth. Inactivation of FOXO3 might therefore be a potential target in treating muscle atrophy [31].

The four mammalian FOXO proteins FOXO1 (FKHR), FOXO3 (FKHRL1), FOXO4 (AFX), and FOXO6 have been identified as the orthologues to DAF-16, the mediator of insulin signaling in *C. elegans*. All of these are part of the forkhead transcription factor family, a family of more than 100 members in different species. These share a certain highly conserved winged helix DNA-binding domain, consisting of 100 nucleotides [32,33,34].

As a downstream member of the IGF-1 signaling pathway, FOXO3 holds a leading position in protein degradation and anabolic/catabolic protein homeostasis within the musculoskeletal system [31]. Through mediation via the PI3K/AKT pathway, lower levels of Insulin-like Growth Factor 1 (IGF-1) lead to a higher activity of FOXO3. The resulting target gene is the ubiquitin ligase *Atrogin-1*, which causes skeletal muscle atrophy [35].

It has been shown that sarcopenia is associated with a higher level of systemic inflammation and a lower level of anabolic hormones indicated by an increase in Tumor Necrosis Factor-Alpha (TNF-α) and a decrease in IGF-1 in elderly humans (≥60 years) [36].

As shown in a mouse model to identify and differentiate age- or caloric-dependent patterns, IGF-1 as an upstream mediator undergoes an age-dependent regulation. Both age-dependent decreases in plasma IGF and, in general, lower IGF plasma levels were observed in calorie-restricted mice [37]. In contrast, Furuyama et al. investigated *Foxo3* expression in rat skeletal muscle and could not verify a change between young and aged rats [38]. This suggests that it is not the overall expression, but rather the regulation of FOXO3 that might cause sarcopenia [35]. It should be noted that another study found no decrease in the PI3K/AKT pathway during aging, and sarcopenia was not caused by FoxO activation. Conversely, overexpression of Akt with suppression of Atrogin-1 resulted in adverse effects including impaired muscle strength and reduced lifespan [39]. However, the current literature is controversial; some authors state beneficial effects on muscle morphology due to *Foxo3* knockdown [40,41,42,43,44,45,46,47]. The latter will be discussed in the following paragraphs, divided by Akt-dependent and -independent approaches.

## 5. PI3K/AKT Pathway

In general, an upregulation of the PI3K/AKT pathway results in inactivation of FOXO3 by phosphorylation. This is governed by IGF-1 as an activator of growth factor receptor protein tyrosine kinases in anabolic situations. In catabolic situations, the absence of IGF-1 results in active FOXO3 that enters the nucleus to bind to the promotor region of the ubiquitin ligases *Atrogin-1* and *Murf-1* [35,48,49,50] (Figure 1). Both (ATROGIN-1 and MURF-1) are E3 ubiquitin ligases that cause polyubiquitination of proteins and result in proteasomal degradation [51]. Autophosphorylation of an activated growth factor receptor protein tyrosine kinase, located at the cell membrane, results in recruiting phosphoinositide 3-kinase (PI3K). This kinase converts phosphatidylinositol-4,5-bisphosphate (PIP2), a membrane-bound molecule, to phosphatidylinositol-3,4,5-trisphosphate (PIP3). Because of its ability to bind signaling molecules containing a certain pleckstrin homology, PIP3 brings both the serine–threonine protein kinase (AKT) and the phosphoinositide-dependent kinase 1 (PDK1) together. If PDK1 is physically close to its downstream target AKT, AKT is activated via phosphorylation. Activated AKT phosphorylates and prevents FOXO3 from entering the nucleus [52] (Figure 1). More specifically, AKT phosphorylates FOXO3 at Ser253, S315, and T32 in vitro and in vivo. The phosphorylated FOXO3 is bound by the protein 14-3-3, which isolates FOXO3 as inactive in the cytoplasm [53]. Notably, there are other proteins that regulate *Foxo3* both negatively and positively (see [54]). Two examples of positive regulation are demonstrated: At first, MAPK-activated protein kinase 5 can phosphorylate and activate FOXO3 in response to DNA damage [55]. Also, the AMP-activated protein kinase activates the transcriptional activity of FOXO3 through phosphorylation at low energy levels [56]. By comparing the contrary regulatory mechanisms of FOXO3, the targetability of *Foxo3* itself is highlighted.

## 6. Post-Translational Modification of FOXO3: Acetylation and Deacetylation

Another regulatory mechanism of FOXO3 involves post-translational modification through acetylation and corresponding deacetylation. The CBP/p300 coactivator mediates acetylation, while SIRT1 and SIRT2 mediate deacetylation [57]. However, acetylation of FOXO3 has been reported to induce its cytosolic translocation and consequent proteasomal degradation in C57BL/6J mice in vivo [58].

On the other hand, it was found that SIRT1-mediated deacetylation of FOXO3 increases its activity in the nucleus accumbens of C57BL/6J mice in vivo [59]. Remarkably, SIRT1-mediated deacetylation of FOXO3 also leads to a decrease in its activity in vitro [60]. Another study confirmed that FOXO3 activity was reduced by SIRT1 deacetylation as well as by SIRT2 in vitro [61]

In conclusion, the post-translational modification of FOXO3 appeared to be intricate. Although its acetylation seemed to reduce its activity, deacetylation has been observed to both increase and decrease FOXO3 activity [57].

## 7. De Novo Protein Synthesis via mTOR Signaling

In vitro studies revealed that IGF-1 caused myotube hypertrophy in C2C12 myoblasts (an immortalized myoblast cell line used to study myogenesis) to be mediated via the PI3K/AKT/mTOR pathway, which was prevented by applying Rapamycin, an inhibitor of the mammalian target of rapamycin (mTOR) [62]. The protein kinase mTOR is contained in two complexes with different functions. The first one, mTORC1, mediates cell growth and is sensitive to rapamycin. The other one, mTORC2, mediates cell survival and proliferation but is insensitive to rapamycin [63]. Increased ATP or amino acid levels, predominantly in anabolic conditions, positively regulate the activity of mTORC1 independently [64,65]. Activated mTORC1 binds to eIF3 and phosphorylates S6K1 and 4E-BP1, leading to an assembly of the preinitiation complex and therefore de novo protein synthesis in vitro [66,67]. Another positive but indirect regulator of mTORC1 is AKT itself. AKT phosphorylates the tuberous sclerosis complex (TSC), leading to a release from the GTPase Rheb which in turn activates mTORC1 [68]. In conclusion, a positive effect of mTORC1 in terms of myotube hypertrophy and protein synthesis is described. In contrast, an in vivo mouse model revealed that mTORC1 inhibition by rapamycin had positive effects on age-related muscle loss, whereas TSC knockout mice (higher levels of active mTORC1) showed a sarcopenic muscle fiber pattern due to impaired stability of the neuromuscular junction [69]. This implies that in addition to atrophy, the etiology of denervation plays an important role in sarcopenia. However, mTORC2 mediates the IGF-1 signaling as a direct target of AKT. Moreover, PDK1 phosphorylates and activates AKT, yet activates mTORC2 by phosphorylation. Via a positive feedback loop, the activation of AKT is boosted by phosphorylation of mTORC2 [70,71].

### Fiber Type Composition during Aging

In the aging of skeletal muscle, different types of fibers follow different paths, which is important to understand for the development of a specific molecular therapy. A comparative study analyzing human M. vastus lateralis biopsies of younger (23–31 years) and older (68–70 years) men identified myosin heavy chain (MHC) type I fibers as constant in size upon aging, but MHC type IIa and IIx fibers decreasing in size in older subjects [72]. The phenomenon of a type II fiber size decrease in aging was identified to be reversed when resistance training (RT) was applied. As a result of 6 months of training, type II fiber size was increased by RT, but type I fibers remained constant in size in younger (23 years) compared to older (71 years) men [73]. The different muscle types were also rigorously studied in a mouse model. Fast-twitch gastrocnemius muscle (type II) was identified to undergo the highest impairment by aging-dependent atrophy, while slow-twitch (type I) soleus muscle remained unaffected. This finding mirrors the observed human fiber type II atrophy due to aging, as described above [74]. As another response to exercise, the peroxisome proliferator-activated receptor γ coactivator 1α (PGC-1α) is more highly expressed in human and rat muscle in in vivo models after endurance training [75,76,77].

Interestingly, PGC-1α transgenic mice are protected from denervation and fasting-induced muscle atrophy and show a reduced expression of genes required for glycolysis and oxidative phosphorylation [78]. This suggests that PGC-1α creates a milieu typically preferred by type I fibers. Consequently, if the PGC-1α gene was placed downstream of the muscle creatine kinase and subsequently selectively expressed in skeletal and cardiac muscle, type II muscle fibers switched and became type I fibers, indicated by the expression of typical genes for type I fibers (such as troponin I, myoglobin, and cytochrome C) and showed a higher fatigue resistance, induced by electrical stimulation [79,80]. Interestingly, PGC-1α is able to block FOXO3 binding to its responsive element on the *Atrogin-1* promotor [78]. Both ubiquitin ligases (*Atrogin-1* and *Murf-1*) have been identified to be expressed more highly and selectively in cardiac and skeletal muscle during muscle atrophy in an in vivo mouse model [81]. Further, skeletal muscle atrophy was observed to be more severe in glycolytic (IId/x and IIb) compared to oxidative (I and IIa) fibers in vivo [82]. These findings suggest a link between FOXO3 and its target genes (*Atrogin-1* and *Murf-1*) and therefore the PI3K/AKT pathway to sarcopenia, highlighting FOXO3 as a possible target.

## 8. The Influence of Physical Activity and Aging upon the Expressional Profile of FOXO3 in Humans

*Foxo3* content and its target genes have also been studied in the context of physical activity. Here, in mice with unilaterally immobilized hindlimbs, immobilization caused an increase in FOXO3 acetylation and a reduction in gastrocnemius muscle weight in vivo. However, this increase was reversible. Within a few days of unrestricted movement, FOXO3 acetylation levels decreased again. The authors conclude that acetylation of FOXO3 promotes muscle atrophy, while deacetylation of FOXO3 increases muscle regeneration potential, highlighting the importance of physical activity [19].

Two in vivo human studies of younger (20 years) and older (70 years) healthy male and female subjects independently showed that *ATROGIN1* and *MURF1* expression did not change during aging [83,84]. Further investigations showed that ubiquitin expression did not change in human rectus abdominis muscles either [85]. However, the overall ubiquitin protein content showed an age-dependent increase. Ubiquitin protein levels were increased in older (70–79 years) human quadriceps muscle biopsies compared to younger (20–29 years) subjects’ biopsies [86]. In contrast, another study identified increased levels of *FOXO3* and *MURF1* (but not *ATROGIN1*) in older healthy females (85 years) compared to younger females (23 years) in M. vastus lateralis biopsies. After a single session of RT, the *FOXO3* expression remained unchanged, whereas *ATROGIN1* expression was markedly increased in older subjects, and *MURF1* accumulated in both groups [87]. A follow-up study in older women (70 years) performing long-term training (12 weeks on a cycle ergometer) revealed decreased *FOXO3* expression levels with no significant effect on the protein levels and no change in expression of *ATROGIN1* and *MURF1* [88] *(*Table 1). An additional comparative study investigated 12 weeks of RT with a focus on FOXO3 protein content in younger (24 years) and older (85 years) females. In the untrained state, older subjects showed lower levels of cytosolic phosphorylated FOXO3 (P-FOXO3) and therefore less inactivated FOXO3. After the training period, increased levels of total nuclear FOXO3 were observed in older subjects. On the other side, younger subjects showed higher levels of P-FOXO3 in response to RT. These results indicate an increase in total nuclear FOXO3 due to aging and impaired nuclear phosphorylation and thus inactivation in response to resistance training, which may attenuate the beneficial effect of physical activity [89] (Table 1).

In summary, these results suggest that the aging muscle is atrophic through increased protein content of Ubiquitin and nuclear FOXO3. Moreover, it suggests an imbalanced anabolic/catabolic interaction, linking FOXO3 as a potential molecular target for the clinical treatment of sarcopenia.

### Therapeutical Targets

To evaluate the targetability of the PI3K/AKT pathway, the current prospects can be subdivided into an AKT-dependent and an AKT-independent treatment strategy.

## 9. AKT-Dependent Manipulation

The upregulation of the PI3K/AKT pathway leads to phosphorylation and inactivation of FOXO3 with an AKT-dependent manipulation of FOXO3 [48]. As nutritional supplementation, L-carnitine treatment was applied in cachectic mice resulting in increased protein levels of phosphorylated FOXO3 and decreased levels of ATROGIN-1 and MURF1 in vivo, suggesting its mediation via the PI3K/AKT pathway and resulting in more active AKT as a potential target in sarcopenia treatment (Figure 2). In addition, the application of L-carnitine increased the cross-sectional area (CSA) and weight of gastrocnemius muscles in cachectic mice in vivo [46]. It has been shown that lower L-carnitine levels are correlated with sarcopenia [90]. However, in elderly women (65–70 years), isolated L-carnitine supplementation was insufficient to increase muscle strength [91]. But, in combination with L-leucine, creatine, and Vitamin D3, an effect could be observed due to an activation of the mTOR pathway, resulting in overall new protein synthesis as described above in healthy adults (55–70 years) [92]. Ubiquitin-specific protease 1 (USP1) acts as a regulator of AKT by inhibiting its function (Figure 2). For AKT activation via phosphorylation, a prior ubiquitination is necessary. As an AKT regulator, USP1 deubiquitinates AKT. Therefore, USP1 indirectly causes less phosphorylation and consequently less activation of AKT. Hence, a knockdown of USP1 in mice resulted in phosphorylated FOXO3 that ameliorated muscle atrophy [93]. In vivo investigations in transgenic mouse cardiomyocytes, expressing constitutively active AKT, revealed a decrease in *Atrogin-1* expression [94]. In summary, these results suggest the activation of AKT to suppress transcriptional atrophy via inactivation of FOXO3. A similar effect was reported for miR-1290 [47] (Figure 2). Briefly, micro-RNAs are small noncoding RNAs that act as post-transcriptional regulators by interacting with mRNA, resulting in a translational blockade or mRNA degradation [95]. MiR-1290 has been negatively correlated with muscle atrophy and was reported to have protective effects in muscle atrophy by inactivation of FOXO3, leading to lower expression of its target genes ATROGIN-1 and MURF1 in vitro. In addition, phenotypical analysis revealed an increase in C2C12 myotube area in vitro and a greater muscle fiber cross-sectional area in a rat muscle atrophy model in vivo. Intriguingly, the combined application of an AKT inhibitor and miR-1290 lacked the positive effects on increased myotube area in vitro, indicating its dependency on AKT [47]. In addition to the AKT-dependent FOXO3 inhibitors, miR-1290 builds a bridge to specific AKT-independent FOXO3 inhibitors, which will be discussed in the following paragraph.

## 10. AKT-Independent Manipulation

AKT-independent targeting acts via selective repression of *Foxo3*. Transfecting an siRNA against *Foxo3* as a post-transcriptional regulation resulted in a decrease in ATROGIN-1 and MURF-1 in L6 rat skeletal muscle in vitro, but the specific *Foxo3* knockdown effects on the phenotype have not been evaluated yet and are not suggested to be long-term due to physiological siRNA degradation [40]. Denervation-induced muscle atrophy, as another etiology of muscle atrophy, was studied in triple Foxo1,3,4-knockout mice, but the molecular effects seem to be transferable. These mice had increased M. soleus CSA, which generated higher contraction forces ex vivo. The same working group generated isolated *Foxo1*-, *Foxo3*-*,* and *Foxo4*-knockout mice. Interestingly, only *Foxo3*-knockout mice were protected from denervation-induced atrophy. These mice showed increased M. gastrocnemius CSA and the impact was primarily on oxidative fibers [41]. These results were also demonstrated in another paper. Here, a triple Foxo1,3,4 knockout prevented both muscle loss and the induction of *Atrogin1* and *Murf1* in the unloaded gastrocnemius and tibialis anterior muscle of the hindlimb in vivo [42]. Another mechanism of AKT-independent manipulation is by the transduction of a dominant negative (DN) FOXO3 protein, the DNA-binding domain of *Foxo3*, suppressing its target gene transcription. A transduction leads to long-term effects, but due to conservations in the DNA binding domain of *Foxo3* (85% identical to *Foxo1* and 75% to *Foxo4* in the mouse), a transduction lacks specificity for *Foxo3* target genes. However, in this model, a knockdown of the downstream target genes (*Atrogin1* and *Murf1*) was observed. The resulting phenotype was characterized by an increase in fiber cross-sectional area in M. tibialis anterior (TA) and M. extensor digitorum in a cancer cachexia mouse model [43]. Further investigations revealed that *Foxo3* was able to increase *Foxo1* and *Foxo4* expression in human fibroblasts in vitro. Moreover, a Foxo binding site was identified within the *Foxo1* promotor in HEK cells in vitro, and further in silico analyses identified it to be located 370 bp before the first exon within the *Foxo1* promotor [44]. The injection of a DN-Foxo plasmid into murine TA and M. soleus decreased *Atrogin-1* and *Murf-1* mRNA content significantly. Moreover, both muscles also showed a phenotypical increase in fiber CSA. Interestingly, suppression of *Foxo* expression resulted in satellite cell proliferation and myofiber fusion in mice in vivo, indicating the relationship between muscle atrophy/growth and satellite cells [45]. Transfection of an siRNA against *Foxo3* before myogenic differentiation repressed the differentiation of myoblasts into myotubes in a *Myod1*-dependent manner in C2C12 myoblasts in vitro. Additionally, the myotubes were again formed by re-expressing *Myod1* within the myoblasts [96]. Similar results were observed by transduction of an siRNA prior to myogenic differentiation. Here, smaller myotubes were observed due to the FOXO3 knockdown. Further, within the first days of differentiation, lower levels of *Atrogin-1* were observed with an increase during later differentiation stages, suggesting a compensatory regulation via *Foxo1* in vitro [97]. These conflicting results imply a negative impact of a *Foxo3* knockdown prior to myogenic differentiation.

### 10.1. Satellite Cells

Satellite cells (SCs) as muscle stem cells are of paramount importance for maintaining muscle mass and regeneration potential by remaining in a post-mitotic, quiescent state until recruitment for differentiation. After a (non-specific) trigger, such as injury, SCs proliferate and form de novo myotubes or fuse to existing myotubes as part of the regeneration process [98,99]. This behavior was studied in SC-ablated mouse hindlimbs, where single myofibers were grafted into hindlimbs. A low number of SCs was sufficient to proliferate, repopulate, and form de novo myofibers in vivo [100]. In the pathologic mechanism of developing sarcopenia, a decline in satellite cell function plays an important role, leading to impaired reversal of muscle atrophy in aging [101]. This link between the SC pool and aging was observed in human skeletal muscle biopsies from young (18–49 years), older (50–69), and senescent (70–86 years) men. In these men, the muscle fiber composition changed, and the number of satellite cells underwent a significant decline during aging. Again, the level of slow oxidative type I fibers was constant, whereas the fast glycolytic type II fibers decreased during aging [102] (Figure 3). This effect was intensified by skeletal muscle atrophy being more severe in type II muscle fibers [82]. In addition to overall atrophy, the number of satellite cells per fiber decreased (Figure 3). Apparently, both trends could be reversed by resistance training in older subjects [82,102].

This effect was similarly studied in a mouse model, where mice performed a 4-week treadmill training, resulting in hypertrophy and promotion of SC proliferation and differentiation. Simultaneously, insulin-like growth factor binding protein 7 (*Igfbp7*) expression was upregulated to suppress the PI3K/AKT pathway and prevent SC exhaustion. In line with that, less mTOR activity led to decreased protein synthesis [103] and apparently less inactivation of FOXO3 can be assumed due to its mediation via the PI3K/AKT pathway [52]. An in vitro study revealed that *Foxo3* overexpression reduces the proliferation of muscle precursor cells [104]. Indicating an important role for *Foxo3* in stem cell homeostasis, another working group revealed FOXO3 protein levels to be higher in quiescent SCs compared to activated SCs. But in response to muscle injury, the number of self-renewed SCs was reduced in *Foxo3*-deleted mice in vivo, pointing out its role in regeneration and long-term maintenance of muscle mass and strength [105]. Further, if the SC pool was subdivided into genuine SCs (preserved during aging) and primed SCs (primed to differentiate), a Foxo activation led to the conversion to genuine SCs while its inactivation favored the primed state [106]. To conclude, *Foxo3* suppression started differentiation, while *Foxo3* activation preserved the stem cell fate, giving *Foxo3* a protective role in stem cell regeneration. One explanation for its protective role might be the ability to promote Notch signaling, which has been shown to be essential for satellite cells to remain quiescent [105,107].

As pointed out in several clinical trials, exercise and resistance training promote muscle hypertrophy. The impact of long-term training was assessed in a prospective study: Older men (74 ± 8 years) performed a 12-week exercise program, resulting in increased muscle strength and increased SC content, quantified via immunohistochemistry, in both type I and type II skeletal muscle fibers [108]. To further characterize the training effects of a single session of high-force eccentric exercise, younger male subjects (23 ± 1 years) performed 300 high-force eccentric knee actions resulting in increased SC content in type II, but interestingly not in type I muscle fibers [109]. Another study compared younger (23–35 years) with older (60–75 years) subjects performing a single training session of eccentric knee actions. Both groups showed significantly increased levels of SCs, but the effect was even higher in younger subjects compared to older subjects. These results suggest that type II fibers are more sensitive to acute stimulation, while type I fibers are activated by regular stimulation. However, both fiber types are characterized by attenuated reactivity in older age [110].

### 10.2. Sarcopenia and Inflammation

Inflammaging, the process of age-associated systemic and chronic low-grade inflammation, is another key player in the pathogenesis of sarcopenia [111] (Figure 4). Its effect was studied in an age-independent inflammation model in vivo: (Nuclear Factor Kappa B Subunit 1) *Nfkb1*-knockout mice (lacking the subunits p105 and p50) showed systemic low-grade inflammation. They were parasymbiontically connected to the systemic circulation of healthy wildtype mice. Six weeks after parabiosis, the wildtype mice showed an increased expression of inflammatory mediators in the bone marrow, such as Interleukin (IL) 1a, IL1b, and TNF-α [112] (Figure 4). Interestingly, all these molecules are also members of the senescence-associated secretory phenotype (SASP) [113]. Increased TNF-α plasma levels were also observed in sarcopenic patients (≥60 years) linking inflammaging to sarcopenia [36]. As a possible explanation, in vitro models showed inhibiting effects of TNF-α on myogenic differentiation [36,114,115]. Further in vitro investigations in C2C12 myotubes identified TNF-α to induce *Atrogin-1* expression independently from the PI3K/AKT pathway via FOXO4, but there is uncertainty regarding the effect on muscle cell morphology because of the overall lower *Foxo4* transcriptomic levels compared to *Foxo1* and *Foxo3* [116]. FOXO3 has anti-inflammatory functions and promotes apoptosis [117,118]. It increases kB-RAS1 protein levels, an inhibitor of NFkB, and the activity of C-Jun N-terminal kinases (JNKs) and has been shown to switch TNF-α signaling towards JNK in human umbilical vein cells in vitro. By promoting the JNK axis, FOXO3 favors apoptosis [118]. Activated JNK inhibits Bcl-2 in the mitochondria directly and indirectly via the Bid and Bax pathways to release cytochrome C (cytC) from the mitochondria, causing apoptosis [119,120]. Interestingly, at 8, 18, 29, and 37 months of age, a study found an increase in apoptosis in the aging rat gastrocnemius muscle. The results showed an aging-dependent increase in Bid, Bax, and Bcl-2 at the protein level with no change in the ratio of Bax/Bcl-2 (ratio of pro- to antiapoptotic). They concluded the increase in apoptosis upon aging to be caspase independent [121]. However, this finding does not necessarily exclude reduced apoptosis mediated through a FOXO3 knockdown in aging (Figure 4). Indeed, *Foxo3*-knockout mice showed increased *Nfkb* expression, leading to T cell hyperproliferation and associated inflammation in vivo closing the loop of inflammation as a driver in sarcopenia [111,117].

### 10.3. Sarcopenia Treatment on the Molecular Level

The data so far seem to pitch a *Foxo3* knockdown as a new target for sarcopenia treatment. Such a concept would require strategies for long-term gene silencing and concepts of its targeted delivery.

## 11. Brief Overview on Available Vectors: AAVs and Lentiviruses

Adeno-associated viruses (AAV) belong to the family of parvoviruses. Their genome contains three genes: *Rep* (Replication), *Cap* (Capsid), and *aap* (Assembly). As vectors in gene delivery, recombinant AAVs (rAAV) are generated by replacing the *Rep* and *Cap* gene with a gene of interest [122,123].

The long-term potential of AAV-mediated gene expression was illustrated in nonhuman primates. In this study, a single intramuscular injection of AAV carrying a gene for erythropoietin was applied. Surprisingly, the expression was stable and long lasting for at least six years in vivo [124]. Apart from the time aspect, the transduction efficiency in muscle fibers is also of crucial importance. Several serotypes (AAV2/1, AAV2/2, AAV2/5, AAV2/7, AAV2/8) were tested and found to be sufficient; in particular, AAV2/1, AAV2/7 and AAV2/8 are to be highlighted. Moreover, they all transduced both fast and slow muscle fibers at the same frequency [125]. Additionally, AAV2/8 transduction in C57/BL6 mice in vivo resulted in a stable transgene expression with only a weak induction of T-cell activation. Further investigation in vitro revealed only a poor transduction of antigen-presenting cells (APCs) leading to poor presentation on major histocompatibility complex I (MHCI) and therefore avoidance of an immune reaction [126].

Lentiviruses belong to the family of retroviruses. The three genes gag (structural proteins), pol (reverse transcriptase), and env (viral envelope) are contained in their genome [127]. Their therapeutic potential was demonstrated in an 18-year-old male patient suffering from beta-thalassemia. Successful ß-globin expression for 21 months (observational ending), mediated by a lentiviral vector and *HMGA2* gene activation, led to an independence from blood transfusion [128]. A striking risk of lentiviral delivery is the oncogenic potential by insertional mutagenesis driven by their integrating behavior [129].

Both AAV (250 trials) and lentiviruses (315 trials) are used in several clinical trials (as of 2021) [130].

## 12. Gene-Silencing Strategies

One gene-silencing strategy is RNA interference (RNAi). As a post-transcriptional regulation, small interfering RNA (siRNA), together with the RNA-induced silencing complex (RISC), degrades mRNA if bound complementarily [131]. To test its in vivo applicability, this concept was applied in a mouse model to demonstrate its gene knockdown potential. RNA interference, degrading the mRNA post-transcriptionally, and U1 interference, inhibiting polyadenylation of pre-mRNA, were used in combination with inhibiting firefly luciferase in mice hindlimb muscle in vivo. For an observational time of 8 weeks, a stable knockdown was observed, indicating its potential for gene silencing [132] (Figure 5).

Another promising strategy is the use of clustered regularly interspaced short palindromic repeats (CRISPR) and the CRISPR-associated endonuclease 9 (CAS9) enzyme. CRISPR RNA (crRNA) defines target specificity. Docked to a target, the CAS9 endonuclease cleaves the nucleic acid. For experimental usage, crRNA is replaced by a guide RNA (sgRNA) complementary to a gene of interest. Knockout is generated by reparation failure of CAS9-generated double strand breaks [133]. A proof-of-principle study investigated the gene-editing potential in mice in vivo. In this study, CRISPR/Cas9 was potent enough to edit the *Pcsk9* gene in mice hepatocytes, resulting in decreased plasma PCSK9 levels, but the duration of this effect remains unknown [134].

For future approaches, the use of targeted protein degradation (TPD) is of increasing interest as a post-translational degradation method. The most well-known technology for TPD is PROteolysis TArgeting Chimera (PROTAC). These molecules consist of a head for binding to the protein of interest and an E3-recruiting ligand, connected by a linker. Therefore, the PROTAC binds the protein of interest and delivers it to the proteasomal degradation system [135]. The degradation of androgen receptors in HeLa cells in vitro has been successfully demonstrated by a working group using this concept [136]. Currently, clinical trials feature PROTAC, for example, in treatment of prostatic and breast cancer, exhibiting promising results that necessitate further investigation [137]. It is noteworthy that the technology of TPD does not solely entail PROTAC and proteasomal degradation as further concepts have been derived and expanded from this idea. In addition to proteasomal degradation, TPD has been extended to include lysosomal degradation. To navigate this extensive and intricate field of TPD, we refer to further literature [135,138]

### Potential Pitfalls in a Foxo3-Targeted Therapy

First of all, FOXO3 is shown to maintain a geroprotective role during aging, as silencing of FOXO3 promotes and activation of FOXO3 slows senescence in primate muscle myotubes [139]. As an often-reviewed driver in carcinogenesis, FOXO3 is described as a tumor suppressor gene that was identified to be inactivated by mutations or posttranslational modifications in cancer [140,141]. There are different examples of *Foxo3* knockdown being associated with cancer progression. An in vivo study in transgenic adenocarcinoma of the mouse prostate (TRAMP) mice observed that blocking the activity of FOXO3 resulted in accelerated progression of prostate cancer [142]. Another study verified dephosphorylated FOXO3 (active FOXO3) to suppress breast cancer growth and favor apoptosis in a rat model in vivo [143]. Moreover, FOXO3 represses the DNA methyltransferase 3B (DNMT3B) by interacting with a binding site within the DNMT3B promotor in human lung cancer cell lines (A549 and CL1-5) in vitro. The same working group also revealed that in lung cancer patients, lower levels of FOXO3 and higher levels of DNMT3B were correlated with a poor prognosis for lung cancer [144]. Intriguingly, hypomethylating agents (azacitidine and decitabine) dephosphorylate FOXO3 (in patient-derived AML cells in vitro), increasing *BIM* and *PUMA* expression, driving apoptosis in SKM-1 cells in vitro [145]. Conflictingly, in vitro studies showed that AKT promoted the survival of vascular smooth muscle cells (VSMCs) by inhibiting FOXO3 and GSK3 [146]. This is relevant in plaque stability in atherosclerosis. Increased apoptosis in VSMCs leads to thinned fibrous caps and reduced collagen destabilizing atherosclerotic plaques in an in vivo mouse model [147].

Taken together, these findings indicate potential pitfalls in *Foxo3* knockdown-based gene therapies. An unspecific *Foxo3* knockdown may have protective effects in atherosclerosis. However, higher levels of FOXO3 and its tumor-suppressive function are favorable in senescence and cancer therapy. To avoid these pitfalls, a local and isolated therapy on the skeletal muscle might prevent systemic effects, while preserving the muscle-specific benefit.

## 13. Conclusions

In summary, physical activity and training are key for maintaining muscle mass and strength through aging. However, the clinical applicability, especially in the injured and frail population, is limited. For immobile and sarcopenic patients, further therapeutical options are needed since nutritional supplementation alone only has a limited potency. Here, we summarized the role of *Foxo3* and its molecular effects on skeletal muscle. Elevated protein levels of nuclear FOXO3 have been observed in sarcopenic human subjects, making it a possible therapeutic target. This could be achieved on multiple levels and via different pre- and post-transcriptional strategies. We compared AKT-dependent and -independent approaches in terms of short- and long-term impact. An auspicious therapy would require long lasting effects, for instance via vector-mediated gene silencing. A *Foxo3* knockdown is supposed to promote satellite cell differentiation and form new muscle fibers and/or recruit new muscle fibers and strength. Care must be taken to ensure that this does not lead to an exhaustion of the stem cell pool, impaired inflammation, promoted senescence, or cancer progression, especially as all these aspects might even reinforce sarcopenia and related diseases.

In summary, FOXO3-mediated genetic therapy might enable molecular treatment of sarcopenia, if a potent and muscle-specific form of application can be devised.

## Figures and Tables

**Figure 1 cells-12-02787-f001:**
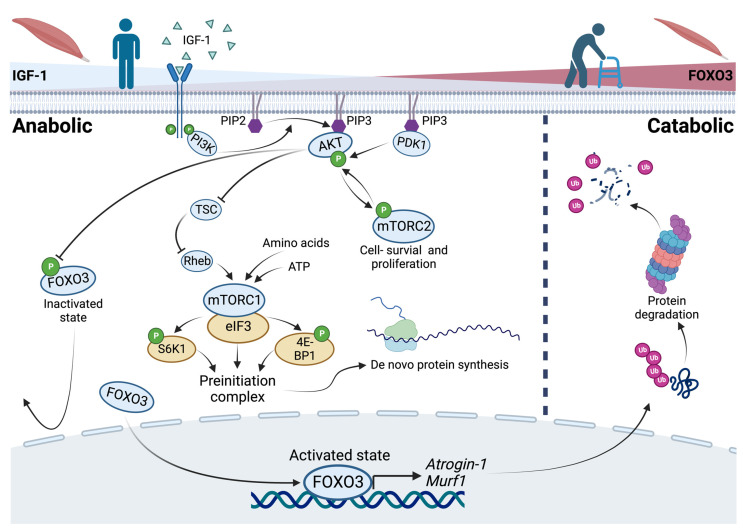
The PI3K/AKT pathway and the role of FOXO3 in anabolic and catabolic conditions as a reflection of changes in the catabolic state, like aging. In anabolic situations, recruitment of phosphoinositide-3-kinases (PI3K) to an activated growth factor receptor results in the conversion of phosphatidylinositol-4,5-bisphosphate (PIP2) to phosphatidylinositol-3,4,5-trisphosphate (PIP3) bringing the serine–threonine protein kinase (AKT) and phosphoinositide-dependent kinase 1 (PDK1) together. AKT inhibits the Forkhead-Box Protein O3 (FOXO3) by phosphorylation. Further, AKT promotes cell survival via the mammalian target of rapamycin complex 2 (mTORC2) and indirectly promotes protein synthesis via the mammalian target of rapamycin complex 1 (mTORC1). In catabolic situations, FOXO3 regulates the expression of its downstream targets *Atrogin-1* and *Murf-1* as ubiquitin ligases that cause proteasomal degradation of proteins (created with BioRender.com).

**Figure 2 cells-12-02787-f002:**
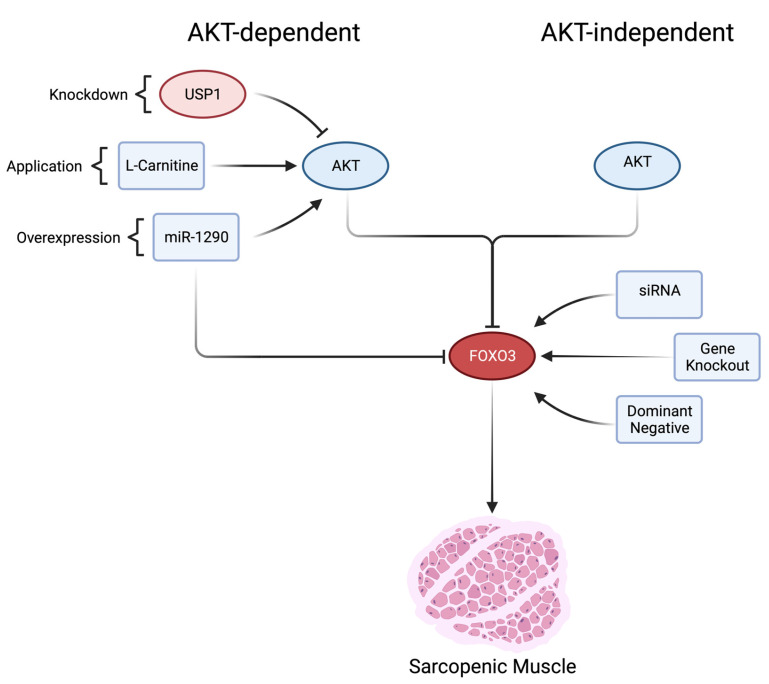
Summary of AKT-dependent and -independent target strategies. In general, AKT-dependent strategies are based on increased activity of AKT. On the contrary, AKT-independent strategies are based on selective repression of FOXO3 (created with BioRender.com).

**Figure 3 cells-12-02787-f003:**
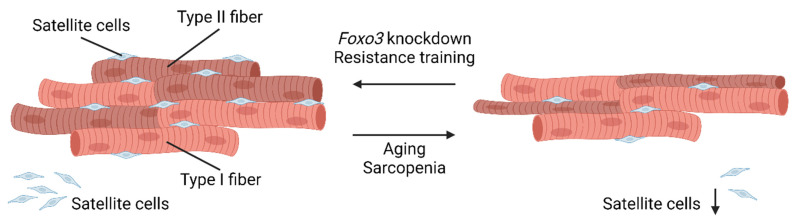
Aging causes skeletal muscle atrophy by mainly affecting type II fibers. During aging, type II fibers and satellite cells are mainly affected, leading to muscle atrophy. As a response to resistance training (RT), satellite cells (SCs) proliferate, regenerate, and form new muscle fibers. Note the reduced amount of satellite cells in the aged muscle (created with BioRender.com).

**Figure 4 cells-12-02787-f004:**
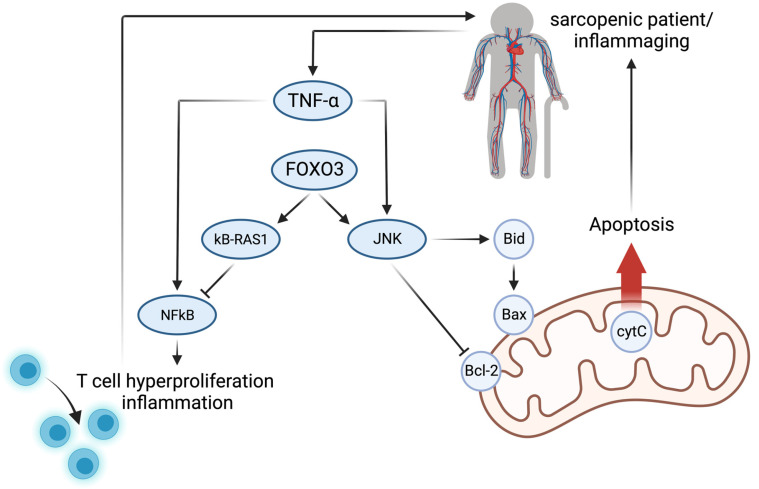
Sarcopenia is linked to inflammation. Increased circulating TNF-α was observed in sarcopenia, leading to inflammation and apoptosis. FOXO3 switches the TNF-α axis from NFkB towards C-Jun N-terminal kinases (JNK), leading to favoring apoptosis. The other way around, a FOXO3 knockdown promotes inflammation (created with BioRender.com).

**Figure 5 cells-12-02787-f005:**
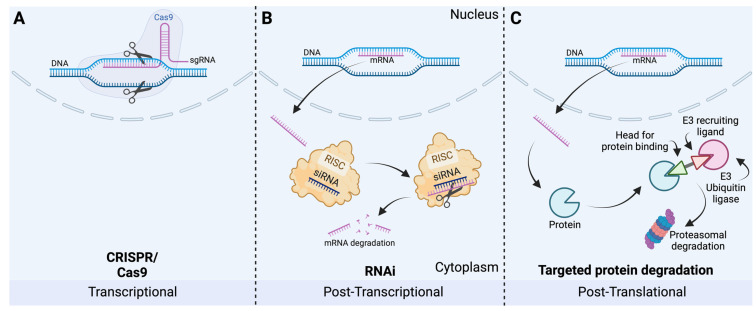
Gene-silencing strategies. (**A**) Clustered regularly interspaced short palindromic repeats (CRISPR)/CRISPR-associated endonuclease 9 (Cas9) directly interacts with the gene at the genomic level. Using a guide RNA (sgRNA), a specific target is silenced directly within the DNA. (**B**) RNA interference (RNAi) as a post-transcriptional regulation interferes with mRNA within the cytoplasm. By forming the RNA-induced silencing complex (RISC), complementary RNA is degraded. (**C**) Targeted protein degradation via PROTAC delivers the protein of interest to the proteasomal degradation machinery (created with BioRender.com).

**Table 1 cells-12-02787-t001:** FOXO3 levels upon different (training) conditions.

FOXO3 Levels	Condition	Model	Reference
*FOXO3* ↓ after 12 weeks on a cycle ergometer in older women	Long-term training	Human	[88]
FOXO3 phosphorylation ↓ before and total nuclear FOXO3 ↑ after 12 weeks of RT in older females	RT	Human	[89]
*FOXO3* ↑ in older healthy females with *FOXO3* expression ↔ after a single session of RT	Aging + RT	Human	[87]
FOXO3 acetylation↑ due to hindlimb immobilization	Immobilization	Mice	[19]
No age-dependent downregulation of the PI3K-AKT pathway	Aging	Mice	[39]

## Data Availability

Not applicable.

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
