# Peer review of "Therapeutic Consequences of Targeting the IGF-1/PI3K/AKT/FOXO3 Axis in Sarcopenia: A Narrative Review"

_cells, 2023, doi:10.3390/cells12242787_

Round 1

Reviewer 1 Report

Comments and Suggestions for Authors

In the present work, the authors perform a narrative review about the Foxo3 as a potential therapeutic target in sarcopenia. In general, the review is well-written with clear sequence. Please, see the minor suggestions as follow.

1)    Title can be improved, since the IGF-1/PI3k/Akt/Foxo3 axis is not the single pathway discussed in the review.

2)    The first topic (“Definitions”) can be changed by “Introduction”.

3)    In the “Epidemiology & economy” section, please provide more recent data.

4)    The figures can be improved. In the Figure 1, Anabolic and Catabolic pathways could be clearly separated. In the Figure 2, for example, Akt-dependent and Akt-independent pathways can be clearly separated and showed in different sides (left and right, for example), but with the same target (FOXO3) in the middle. In the Figure 3, mechanisms involved in aging sarcopenia and resistance training can be added. In the Figure 4, the controversial effects of FOXO3 on inflammation on sarcopenia can be detailed. Do the increased T cell proliferation and apoptosis contribute to the sarcopenia development?

5)    The last section ("Sarcopenia treatment") focuses only in cellular and animal models, based on genetic therapies, with low potential clinical applicability to humans. This section could be greatly improved with discussion addressing FOXO3 as a molecular target of potential strategies of intervention in humans, including nutritional and physical exercise, for example.

Author Response

In the present work, the authors perform a narrative review about the Foxo3 as a potential therapeutic target in sarcopenia. In general, the review is well-written with clear sequence. Please, see the minor suggestions as follow.

1)    Title can be improved, since the IGF-1/PI3k/Akt/Foxo3 axis is not the single pathway discussed in the review.

Indeed, we agree that different pathways were addressed within this review. Nevertheless, our main focus and statements were derived from the idea of targeting the IGF-1/PI3k/Akt/Foxo3 pathway, especially and specific in the skeletal muscle, leading to a decrease of muscle specific ubiquitin ligases Atrogin-1 and Murf1.

2)    The first topic (“Definitions”) can be changed by “Introduction”.

We thank the reviewer for this improvement. The topic is changed to “Introduction” to give the reader a better understanding of the structure of the review.

3)    In the “Epidemiology & economy” section, please provide more recent data.

We thank the reviewer for this comment. More recent data have been added on the prevalence and incidence of sarcopenia (line 90-94) as the association between grip strength and mortality highlighting the importance of parameters used to define sarcopenia and the risk of death (line 103-108)

Among a population of 518 male European participants aged 40-79 years, an incidence of 1.6% was observed after a follow up of 4.3 years, as defined by the EWGSOP criteria for sarcopenia[6]. Furthermore, another study (n=719) conducted on older (≥85years) female and male participants found a prevalence of 21% for sarcopenia based on the EWGSOP criteria with an incidence of 3.7% in a follow up after 3years[7]. [A study investigating sarcopenia in older hospitalized patients (≥65years) revealed 35% of the patients to be sarcopenic at hospital admission (prevalence) and 15% of non-sarcopenic patients to develop sarcopenia until discharge (incidence in hospitalized patients) indicating its relevance and tremendous impact on the healthcare system[8].

A follow-up study in geriatric patients (7.1% sarcopenic according to EWGSOP definition, n=445) observed that during a period of three years, about half of the patients (≥65years) fell one or multiple times, while the fall rate was higher in sarcopenic patients[9]. The increased rate of falling was confirmed in an extensive meta-analysis (n=45,926),] in which a positive correlation of falls with fractures in sarcopenic patients was observed [10]. Intriguingly, grip strength, as a major criterion used to identify sarcopenia, but also walking speed, chair raises and standing balance, were found to be associated with mortality in a meta-analysis. Individuals with lower performance in these metrics were found to have a greater risk of mortality, highlighting the importance of parameters used to identify sarcopenia and risk of death[11]. [In addition, the hospital stay is prolonged in sarcopenic compared to non-sarcopenic patients, which has an economic impact on the health system[12]. In detail, prolonged hospital stays lead to a higher socioeconomic burden in the healthcare system resulting in an estimated additional costs of $40.4 billion for patients with sarcopenia in the US in 2014[13].]

4)    The figures can be improved. In the Figure 1, Anabolic and Catabolic pathways could be clearly separated. In the Figure 2, for example, Akt-dependent and Akt-independent pathways can be clearly separated and showed in different sides (left and right, for example), but with the same target (FOXO3) in the middle. In the Figure 3, mechanisms involved in aging sarcopenia and resistance training can be added. In the Figure 4, the controversial effects of FOXO3 on inflammation on sarcopenia can be detailed. Do the increased T cell proliferation and apoptosis contribute to the sarcopenia development?

We thank the reviewer for this suggestion. Fig.1 has been updated to separate the anabolic and catabolic pathways more stringently. For the Fig.2, we have revised the layout to emphasize Foxo3 as a central component, with clear visual separation of Akt-dependent and Akt-independent approaches. Fig.3 conveys a simplified message of the age-dependent loss of muscle, particularly for type II fibers, and satellite cells. The labelling of the arrows intends to support the understanding of the overall mechanism. Therefore, and for simplicity, we decided not to add training mechanisms. The Fig.4 was updated with inflammation, resulting in closure of the loop between inflammation and apoptosis as drivers of sarcopenia. We further explained the connection of apoptosis and the pathogenesis of sarcopenia in lines 522-530.:

[Inflammaging, the process of an age-associated systemic and chronic low-grade inflammation is another key player in the pathogenesis of sarcopenia[111] (Fig. 4). Its effect was studied in an age-independent inflammation model in vivo: (Nuclear Factor Kappa B Subunit 1) Nfkb1 knockout mice (lacking the subunits p105 and p50) show systemic low-grade inflammation. They were parasymbiontically connected to the systemic circulation of healthy wildtype mice. Six weeks after parabiosis, the wildtype mice showed an increased expression of inflammatory mediators in the bone marrow such as Interleukin (IL) 1a, IL1b and TNF-α[112] (Fig. 4). Interestingly, all these molecules are also members of the senescence-associated secretory phenotype (SASP)[113]. Increased TNF-α plasma levels were also observed in sarcopenic patients (≥60years) linking inflammaging to sarcopenia[36]. As a possible explanation, in vitro models showed an inhibiting effects of TNF-α on myogenic differentiation[36,114,115]. Further in vitro investigations in C2C12 myotubes identified TNF-α to induce the Atrogin-1 expression independently from the PI3K/AKT pathway via FOXO4, but there is uncertainty regarding the effect on muscle cell morphology because of the overall lower Foxo4 transcriptomic levels compared to Foxo1 and Foxo3[116]. FOXO3 has anti-inflammatory functions and promotes apoptosis[117,118]. It increases the kB-RAS1 protein level, an inhibitor of NFkB, and the activity of C-Jun N-terminal kinases (JNKs) and has been shown to switch the TNF-α signaling towards JNK in human umbilical vein cells in vitro. By promoting the JNK axis, FOXO3 is favoring apoptosis[118]. Activated JNK inhibits Bcl-2 at the mitochondria directly and indirectly via the Bid and Bax pathway to release cytochrome C (cytC) from the mitochondria causing apoptosis[119,120].] Interestingly, at 8-, 18-, 29- and 37-months of age, a study found an increase in apoptosis in the aging rat gastrocnemius muscle. The results showed an age-dependent increase in Bid, Bax and Bcl-2 on the protein level with no change in the ratio of Bax/Bcl-2 (ratio of pro-to antiapoptotic). They concluded the increase of apoptosis upon aging to be caspase independent [121]. However, this finding does not necessarily exclude a reduced apoptosis mediated through a FOXO3 knockdown in aging [(Fig. 4). Indeed, Foxo3 knockout mice showed increased Nfkb expression leading to T cell hyperproliferation and associated inflammation in vivo] closing the loop of inflammation as a driver in sarcopenia[111,117].

5)    The last section ("Sarcopenia treatment") focuses only in cellular and animal models, based on genetic therapies, with low potential clinical applicability to humans. This section could be greatly improved with discussion addressing FOXO3 as a molecular target of potential strategies of intervention in humans, including nutritional and physical exercise, for example.

We agree with the reviewer that our last section focused only on genetic therapies. To address interventions in humans we discuss concepts in sarcopenia treatment in the section “Current therapeutic concepts in sarcopenia” (lines 114-170). Following FOXO3 as a molecular target, the influence of physical exercise, as a major and important treatment in sarcopenia, is discussed in “The influence of physical activity and aging upon the expressional profile of FOXO3 in humans” (lines 316-351). Unfortunately, to date there is a lack of studies addressing FOXO3 in the treatment of sarcopenia in humans.

Reviewer 2 Report

Comments and Suggestions for Authors

Sarcopenia is the gradual loss of muscle mass, strength, and function that occurs with aging. It is a natural part of the aging process and can contribute to various health problems, including frailty and an increased risk of falls and fractures. Several factors contribute to sarcopenia, including hormonal changes, decreased physical activity, poor nutrition, and inflammation. As people age, they often experience a decline in the production of hormones such as testosterone and growth hormone, which play a role in maintaining muscle mass. Additionally, reduced physical activity and inadequate nutrition can further exacerbate muscle loss.

Forkhead box O3 (FoxO3) is a transcription factor that is crucial in regulating various cellular processes, including those involved in muscle growth and maintenance. It has been implicated in the aging process and age-related conditions, including sarcopenia. Researchers are exploring the potential of FoxO3 as a target for treatments aimed at preventing or mitigating sarcopenia.

In the first part of the manuscript, the authors described the epidemiology and current therapeutics in sarcopenia. Next, the authors briefly discussed molecular mechanisms of Foxo-3 activation and biological pathways, followed by the potential FoxO3 targeted therapy to treat sarcopenia, which is the primary scope of this manuscript. Considering this aspect, the current version of the manuscript must be improved in the following way:

1) Addition of a table mentioning the FoxO3 level in different physiological and pathological conditions with appropriate references. This will provide a broad overview of the manuscript.

2) The authors must discuss the role of acetylation and deacetylation in FoxO3 activity.

3) The authors should also consider mentioning the role of E3 ligase Fbxo32, which is regulated by FoxO3 (PMID: 37681900). Also, discussion on different gene expressions upon FoxO3 activation is limited. Authors must include it in the manuscript. Earlier study demonstrated that inactivation of FoxO3 is a potential biomarker and a driver for primate skeletal muscle aging (PMID: 36921027). The authors must discuss this article here as well.

4) The authors discussed the AAVs and lentiviruses and gene silencing strategy to downregulate the FoxO3 level. It is also important to talk about the possibility of using targeted protein degradation, as this approach is broadly used in regulating the protein level using cellular ubiquitin-proteasome system.

5) The authors added a short section about the pitfalls of targeting the FoxO3 level for the treatment. The section is largely missing a more critical discussion about how to avoid the pitfalls to increase the effectiveness of this targeted approach.

Author Response

Sarcopenia is the gradual loss of muscle mass, strength, and function that occurs with aging. It is a natural part of the aging process and can contribute to various health problems, including frailty and an increased risk of falls and fractures. Several factors contribute to sarcopenia, including hormonal changes, decreased physical activity, poor nutrition, and inflammation. As people age, they often experience a decline in the production of hormones such as testosterone and growth hormone, which play a role in maintaining muscle mass. Additionally, reduced physical activity and inadequate nutrition can further exacerbate muscle loss.

Forkhead box O3 (FoxO3) is a transcription factor that is crucial in regulating various cellular processes, including those involved in muscle growth and maintenance. It has been implicated in the aging process and age-related conditions, including sarcopenia. Researchers are exploring the potential of FoxO3 as a target for treatments aimed at preventing or mitigating sarcopenia.

In the first part of the manuscript, the authors described the epidemiology and current therapeutics in sarcopenia. Next, the authors briefly discussed molecular mechanisms of Foxo-3 activation and biological pathways, followed by the potential FoxO3 targeted therapy to treat sarcopenia, which is the primary scope of this manuscript. Considering this aspect, the current version of the manuscript must be improved in the following way:

1) Addition of a table mentioning the FoxO3 level in different physiological and pathological conditions with appropriate references. This will provide a broad overview of the manuscript.

We agree with the reviewers point. For an easy overview, we added a table following the paragraph “The influence of physical activity and aging upon the expressional profile of FOXO3 in humans” and emphasize the effect of training/condition on the FOXO3 level (line 353).

Table 1: FOXO3 levels upon different (training) conditions

FOXO3 levels

Condition

Model

Reference

FOXO3

after 12 weeks on a cycle ergometer in older women

Long term training

Human

[88]

FOXO3 phosphorylation ↓ before and

total nuclear FOXO3 ↑

after 12 weeks of RT in older female

RT

Human

[89]

FOXO3 ↑

in older healthy female

with FOXO3 expression ↔ after a single session of RT

Aging + RT

Human

[87]

FOXO3 acetylation ↑

due to hindlimb immobilization

Immobilization

Mice

[19]

No age-dependent downregulation of the PI3K-AKT pathway

Ageing

Mice

[39]

2) The authors must discuss the role of acetylation and deacetylation in FoxO3 activity.

We thank the reviewer for this important comment. Therefore, a new section was established addressing the interesting role of acetylation and deacetylation of FOXO3 (line 229-242). In the following section we provide details on a statement that was only mentioned briefly in the review before, as we consider it to be an excellent complement to the new paragraph on acetylation (line 320-324)

(line 229-242) Post translational modification of FOXO3: Acetylation and Deacetylation

Another regulatory mechanism of FOXO3 involves posttranslational modification through acetylation and corresponding deacetylation. CBP/p300 coactivator mediates acetylation, while SIRT1 and SIRT2 mediate deacetylation[57]. However, acetylation of FOXO3 has been reported to induce its cytosolic translocation and consequent proteasomal degradation in C57BL/6J mice in vivo[58].

On the other hand, it was found that SIRT1-mediated deacetylation of FOXO3 increases its activity in the nucleus accumbens of C57BL/6J mice in vivo[59]. Remarkably, SIRT1-mediated deacetylation of FOXO3 also leads to a decrease in its activity in vitro[60]. Another study confirmed that FOXO3 activity was reduced by SIRT1 deacetylation as well as by SIRT2 in vitro[61]

In conclusion, the post-translational modification of FOXO3 appeared to be intricate. Although its acetylation seemed to reduce its activity, the deacetylation has been observed to both increase and decrease the FOXO3 activity[57].

(line 320-324) Here, in mice with unilaterally immobilized hindlimbs, immobilization caused an increase in FOXO3 acetylation and reduction of gastrocnemius muscle weight in vivo. However, this increase was reversible. Within a few days of unrestricted movement, the FOXO3 acetylation levels decreased again. The authors conclude the acetylation of FOXO3 to promote muscle atrophy, while deacetylation of FOXO3 increases muscles’ regeneration potential, highlighting the importance of physical activity[19].

3) The authors should also consider mentioning the role of E3 ligase Fbxo32, which is regulated by FoxO3 (PMID: 37681900). Also, discussion on different gene expressions upon FoxO3 activation is limited. Authors must include it in the manuscript. Earlier study demonstrated that inactivation of FoxO3 is a potential biomarker and a driver for primate skeletal muscle aging (PMID: 36921027). The authors must discuss this article here as well.

We thank the reviewer for this comment. Indeed, the PMID: 37681900 (here Ref [97]) paper is already cited in this review in the section on “AKT-independent targeting of FOXO3” on the background of the FOXO3 knockdown mediated inhibition of myogenic differentiation (line 424-432). We expanded their statement about Fbxo32, also known as Atrogin-1, and added it to our discussion. The PMID: 36921027 paper is already cited and discussed. We additionally included their statement on the geroprotective role of FOXO3 in the section on “Potential pitfalls in a Foxo3-targeted therapy” (line 616-618) (here Ref [134]).

(line 424-432) Transfection of a siRNA against Foxo3 before myogenic differentiation repressed the differentiation of myoblasts into myotubes in a Myod1 dependent manner in C2C12 myoblasts in vitro. Additionally, the myotubes were again formed by re-expressing Myod1 within the myoblasts[96]. Similar results were observed by transduction of an siRNA prior to myogenic differentiation. Here, smaller myotubes were observed due to the FOXO3 knockdown. Further, within the first days of differentiation, lower levels of Atrogin-1 were observed with an increase during later differentiation stages suggesting a compensatory regulation via Foxo1 in vitro[97].

(line 616-618) First of all, FOXO3 is shown to maintain a geroprotective role during aging, as silencing of FOXO3 promotes, and activation of FOXO3 slows senescence in primate muscle myotubes [134].”

4) The authors discussed the AAVs and lentiviruses and gene silencing strategy to downregulate the FoxO3 level. It is also important to talk about the possibility of using targeted protein degradation, as this approach is broadly used in regulating the protein level using cellular ubiquitin-proteasome system.

We agree with reviewer that targeted protein degradation is an important protein regulatory mechanism and thank for this observation. Following RNAi and CRISPR we expanded the section of “Gene silencing strategies” to targeted protein degradation (lines 590-602). We also included this silencing strategy in the Fig.5.

For future approaches, the use of targeted protein degradation (TPD) is of increasing interest, as a post-translational degradation method. A well-known technology for TPD is PROteolysis TArgeting Chimera (PROTAC). These molecules consist of a head for binding to the protein of interest and an E3 recruiting ligand, connected by a linker. Therefore, the PROTAC bind the protein of interest and deliver it to the proteasomal degradation system[134]. The degradation of androgen receptor in HeLa cells in vitro has been successfully demonstrated by a working group using this concept[135]. Currently, clinical trials feature PROTAC for example in treatment of prostatic and breast cancer, exhibiting promising results that necessitate further investigation[136]. It is noteworthy that the technology of TPD does not solely entail PROTAC and the proteasomal degradation as further concepts have been derived and expanded from this idea. In addition to proteasomal degradation, TPD has been extended to include lysosomal degradation. To navigate this extensive and intricate field of TPD, we refer to further literature[134,137]

Fig.5: Gene silencing strategies. (A) Clustered regularly interspaced short palindromic repeats (CRISPR)/CRISPR associated endonuclease 9 (Cas9) directly interacts with the gene on the genomic level. Using a guide RNA (sgRNA), a specific target is silenced directly within the DNA. (B) RNA interference (RNAi) as a posttranscriptional regulation interferes with the mRNA within the cytoplasm. By forming the RNA induced silencing complex (RISC), complementary RNA is degraded. (C) Targeted protein degradation via PROTAC delivers the protein of interest to the proteasomal degradation machinery. (Created with BioRender.com)

5) The authors added a short section about the pitfalls of targeting the FoxO3 level for the treatment. The section is largely missing a more critical discussion about how to avoid the pitfalls to increase the effectiveness of this targeted approach.

We thank the reviewer for this comment on the Foxo3 pitfalls in gene therapy. Subsequently, we expanded the summarizing section (line 639-641) of the paragraph “Potential pitfalls in a Foxo3-targeted therapy”. A specific muscle application might limit the systemic side effect. Nevertheless, limiting muscle specific side effects requires further studies.

[Taken together, these findings indicate potential pitfalls in Foxo3 knockdown-based gene therapies. An unspecific Foxo3 knockdown may have protective effects in atherosclerosis. However, higher levels of FOXO3 and its tumor suppressive function are favorable in senescence and cancer therapy.] To avoid the mentioned pitfalls, a local and isolated therapy on the skeletal muscle might prevent systemic effects, but rather not the muscle specific side-effects.
